# Knowledge and Attitudes Concerning Herpes Zoster among People with COPD: An Interventional Survey Study

**DOI:** 10.3390/vaccines10030420

**Published:** 2022-03-10

**Authors:** Barbara P. Yawn, Debora D. Merrill, Sergio Martinez, Elisabeth Callen, Janice Cotton, Dennis Williams, Natalia Y. Loskutova

**Affiliations:** 1COPD Foundation, Miami, FL 33134, USA; dmerrill@copdfoundation.org (D.D.M.); smartinez@copdfoundation.org (S.M.); gentlejan6@yahoo.com (J.C.); 2Department of Family and Community Health, University of Minnesota, Minneapolis, MN 55455, USA; 3American Academy of Family Physicians National Research Network, Leawood, KS 66211, USA; ecallen@aafp.org (E.C.); nloskutova@kumc.edu (N.Y.L.); 4Division of Pharmacotherapy and Experimental Therapeutics, Eshelman School of Pharmacy, University of North Carolina, Chapel Hill, NC 27514, USA; dwilliams@unc.edu

**Keywords:** chronic obstructive pulmonary disease (COPD), herpes zoster, survey, patient perspectives, vaccination, education, vaccine preventable disease

## Abstract

Herpes zoster (HZ) is common in older adults with conditions such as chronic obstructive pulmonary disease (COPD). Effective prevention is available through vaccination, but HZ vaccine uptake remains incomplete. Using an online survey of people with self-reported COPD, ShiPPS assessed HZ risk awareness, HZ vaccine use and barriers, and the impact of an HZ educational video on vaccine intent. USA members of the COPD Foundation’s Patient-Powered Research Network aged >50 years were surveyed in fall 2020. The responses were analyzed using descriptive and comparative statistics. Of the 735 respondents (59.6% female, mean age 68.5 years), 192 (26.1%) reported previous HZ, of whom 49 (25.5%) reported increased COPD symptoms during HZ episodes. Most participants (94.0%) knew of HZ vaccines, but only 33.1% reported receiving the Advisory Committee on Immunization Practices-preferred recombinant HZ vaccination. The recall of receiving HZ vaccine recommendations differed by the site attended: 68.8% primary care, 26.6% pulmonology offices. Most (74.7%) were unaware that COPD increases HZ risk. Among unvaccinated participants, interest in getting the HZ vaccine increased from 32.0% to 73.5% after watching the video. These results highlight the need for people with COPD to receive further HZ education, such as the five-minute video, and HZ vaccine recommendations from healthcare professionals.

## 1. Introduction

Herpes zoster (HZ), or shingles, a painful dermatomal vesicular rash, is caused by the reactivation of a latent varicella-zoster virus (VZV) infection and is increasingly common as adults age, affecting almost one in three people in the United States (USA) during their lifetime [1,2,3,4]. A significant subgroup of people with shingles experience complications, including long-lasting neuropathic pain (postherpetic neuralgia (PHN)), eye complications (herpes zoster ophthalmicus), or recurrences [2,5,6]. The economic burden of HZ and its complications, including direct medical care costs as well as productivity losses, is estimated to be $2.4 billion annually in the USA [7,8,9,10].

The risk factors for shingles include the well-recognized causes of immunosuppression as well as several chronic illnesses, including chronic obstructive pulmonary disease (COPD) [6,11,12,13,14,15,16]. The risk of developing HZ has been reported to be from 50% to over 200% higher in people with COPD versus without COPD [12,13,14,15,16,17,18]. However, little is known about the COPD-related burdens, such as dyspnea and exacerbations, that may occur with HZ in people with COPD [19].

Vaccination for shingles prevention is widely available in the USA and is highly efficacious, with over 90% protection reported [20,21,22,23,24]. The Advisory Committee on Immunization Practices (ACIP) now preferentially recommends the recombinant vaccine to prevent HZ and its related complications in adults aged ≥50 years and in those who have previously received the live attenuated vaccine. The recombinant vaccine is hereafter referred to as the ‘ACIP-preferred’ HZ vaccination [23]. While the uptake of the shingles vaccines has been increasing [25,26], there is little information on current rates of ACIP-preferred HZ vaccination or the extent and reasons for HZ vaccine hesitancy in people with COPD [27,28,29].

To provide these insights, we conducted the Shingles Patient Prevention Study (ShiPPS). The primary objective of this survey study was to assess ACIP-preferred HZ vaccine use and the barriers to vaccination among people with COPD. The secondary objective was to assess knowledge, beliefs, and attitudes regarding HZ and its burden among people with COPD. The exploratory-interventional objective was to assess the impact of a short (five-minute) educational video on the awareness of the HZ risks associated with COPD and the interest in future HZ vaccination among participants who reported not having already completed ACIP-preferred HZ vaccinations.

## 2. Methods

### 2.1. Study Design and Ethics

ShiPPS was a descriptive cross-sectional survey, with an embedded video intervention. The study was completed in two parts within the same population and conducted in the USA. Part 1 aimed to understand the level of HZ awareness and risk, perceived burden, and impact on people with COPD, as well as the level of HZ vaccine awareness, use, and barriers. Participants who reported in Part 1 of the survey that they were unaware of the increased risk of HZ in people with COPD or of the available HZ vaccines, or that they had not initiated the ACIP-preferred two-shot vaccine, were invited to watch a five-minute video (the intervention). After watching the video, participants answered questions in Part 2 of the survey regarding the usefulness of the educational information and their post-video awareness of the risk of HZ in people with COPD, as well as interest in HZ vaccination.

The study was conducted in accordance with good clinical practice guidelines, applicable regulatory requirements, and the principles of the Declaration of Helsinki. The study was approved by the Western Institutional Review Board (WIRB; Puyallup, WA, USA) and the American Academy of Family Physicians (AAFP) IRB. After reading the survey instructions, a participant’s decision to complete the survey was taken as implied consent, as designated by the WIRB and confirmed by the AAFP IRB.

### 2.2. Study Population

Individuals with self-reported COPD who were registrants of the COPD Foundation’s Patient-Powered Research Network (COPD PPRN; WIRB protocol #20141136) and had completed the baseline PPRN survey were invited to participate in this study. The COPD PPRN registry is housed on a secure online platform and has enrolled 5319 adults aged ≥50 years with a self-reported physician diagnosis of COPD. These individuals were invited to participate via email, sent through the COPD PPRN portal. Registry populations may not be representative of the population of interest, since most require active choices to participate and this registry requires access to online registration.

### 2.3. Survey

The online survey included items from published patient surveys [30,31,32,33,34,35], and its development was guided by experts in internal medicine, family medicine, preventive medicine, and pharmacy, as well as people with COPD, using an iterative process. The survey, including the skip patterns of participants’ possible routes through the survey based on conditional responses, is provided in the Appendix A. Part 1 of the survey queried unvaccinated individuals about their level of interest in future HZ vaccines while the post-video survey queried them about future vaccine intent—an intention for action rather than merely a level of interest. The estimated time required to complete the survey was 10–15 min for Part 1 and 8 min for Part 2, including 5 min to view the video (total for Parts 1 and 2: 18–23 min). The survey launched on 21 September 2020, and closed on 10 November 2020 (7-week duration).

### 2.4. Intervention

The online educational video was embedded into the survey administration platform and described the epidemiology and burden of HZ, a patient’s testimony about the impact of HZ while living with COPD, and the opportunities for HZ prevention. A link to the video is provided in Appendix A. The intervention was offered to participants who reported any of the following: had not heard of HZ or available HZ vaccines; did not think COPD increased the risk of HZ; or had not received at least one dose of the ACIP-preferred HZ vaccine [23].

### 2.5. Statistics

For the primary and secondary objectives, responses to questions asked in Part 1 of the survey, before exposure to intervention (educational video), were analyzed using common descriptive statistics. Discrete variables, including binary, multiple-choice, and Likert scale questions, were described using the frequency and percentage or proportion. Means and standard deviations were used to describe continuous variables. For the interventional objective, responses to questions asked in Part 2 of the survey, after intervention with the educational video, were analyzed by simple descriptive statistics (similar to Part 1), as well as by comparing responses to similar Part 1 and Part 2 questions using Chi-squared and analysis of variance statistics (including Welch’s F-Test). To further evaluate the effect of the video, we completed multinomial ordinal regression with logit link function and ordinal association measure gamma assessment. We used a four-level ordinal scale from “not interested, won’t get vaccine” to “will consider vaccine” to “will discuss with HCP” to “plan to get the vaccine”. Participants with missing or incomplete data, i.e., participants who failed to answer a question, were excluded from the analyses of those questions. Based on the descriptive nature of the survey, the feasibility of data collection during the limited period during which the survey was open to participants, and the size of the relevant published survey studies, the estimated target sample size was arbitrarily set at 800 people with COPD. All analyses were conducted using IBM SPSS, Version 25.0 (IBM Corp., Armonk, NY, USA). Bonferroni correction was used to address multiple comparisons for all post-hoc univariate analyses.

## 3. Results

### 3.1. Population Characteristics

Survey invitations were sent to 5319 COPD PPRN members (aged ≥50 years with self-reported COPD). The number of respondents was 771 (response rate of 14.5%) (Figure 1); of these, 735 respondents completed Part 1 of the survey and were included in the analyses. The participants’ average age was 68.5 ± 7.9 years, 438 (59.6%) participants were female, and 395 (53.7%) reported having one or more COPD exacerbations in the prior 12 months (Table 1). These demographic characteristics are similar to those for the entire COPD PPRN population: mean age 68.1 (standard deviation 8.8) years, 60.1% females, and 50.3% reported exacerbations in the prior 12 months. 

Half (*n* = 389, 52.9%) of participants rated their overall heath as poor or fair. Among the 735 participants, 706 (96.1%) had heard of HZ and 192 (26.1%) reported one or more previous episodes of HZ. Nearly all (*n* = 691, 94.0%) knew that vaccine prevention for HZ is available, but only 419 (57.0%) reported having received any HZ vaccine(s). Overall, 243 (33.1%) reported having received the ACIP-preferred HZ vaccination [23], while another 60 (8.2%) reported that they were unsure which HZ vaccine(s) or how many HZ vaccine shots they had received.

### 3.2. Experience and Perceived Burden of HZ

Of the 192 respondents (26.1% of total) who reported a prior history of shingles, more were women than men (29% vs. 22%, *p* = 0.035) and cases were more common among those reporting 15 or more days of poor physical health per month in the past year (*p* = 0.006). The respondents universally reported that the HZ episode included a rash and pain or itching, of which 20.8% said those symptoms were not bothersome and another 51.0% said that they were bothersome for two weeks or less. Most, 98.3%, sought health care services for the episode, with 74.5% reporting that they received prescription therapy and 10.9% reporting an HZ-related hospitalization, more common in men than women (*p* = 0.009). Figure 2 highlights the HZ symptoms reported to last >2 weeks, the complications including COPD-related issues that the patients associated with the acute HZ episode, and the HZ burden on activities for >2 weeks. The COPD or respiratory symptoms which the patients felt were related to the acute HZ episode are of special interest since these are rarely assessed or reported.

In this group of respondents, HZ episodes were associated with increased COPD symptoms or dyspnea in 49 (25.5%) participants, and a COPD exacerbation, or “flare”, in 23 (12.0%) participants. Higher rates of increased COPD symptoms, dyspnea, and exacerbations were apparent in individuals who reported more days of poor physical health, poor mental health, or inability to perform usual activities (>15 days/month), or those individuals having had two or more COPD exacerbations in the prior 12 months (*p* < 0.05 for all; except COPD symptoms and poor mental health, *p* = 0.06) (Appendix A). The highest reported rate of HZ-associated COPD exacerbation was among those aged 50–59 years (*p* = 0.005).

Whether or not participants (*n* = 735) had personally experienced HZ, most (*n* = 492, 69.7%) knew others who had experienced HZ. When considering their own experience of HZ (Table 2) and that of people they knew (Appendix A), participants reported high rates of perceived HZ impact related to pain and missing work or activities. Most participants reported a perceived importance of HZ prevention, including to prevent the HZ-related increase in COPD symptoms (Table 2).

### 3.3. Perceived HZ Risk Associated with COPD and Opportunities for HZ Prevention

Regarding participants’ perceived risk of getting HZ, 388 (52.8%) of the 735 participants believed their risk to be the same or lower than other adults without COPD (*n* = 266, 36.2% same risk; *n* = 122, 16.6% lower risk), while only 186 (25.3%) correctly reported their perceived risk to be greater and 161 (21.9%) were unsure of the relative risk.

When asked if HZ vaccination had been recommended at the healthcare sites they attended, 68.8% (*n* = 425/618) of participants reported that they learned about the vaccine at a primary care office, while 26.6% (*n* = 149/560) reported having received information at a pulmonologist visit, and 48.5% (*n* = 283/584) when going to a pharmacy. As opposed to the higher primary care recommendation rates reported in all older groups, the youngest age group (50–59 years) reported the lowest rate of having received information from primary care (*p* < 0.001) but the highest rate from pulmonology offices (*p* = 0.028). Participant-reported rates of discussion or recommendations by an HCP for influenza and pneumococcal vaccines were substantially higher than for HZ vaccines (influenza: primary care, 94.4%; pulmonologists, 75.1%; and “pneumonia”: primary care, 88.7%; pulmonologists, 71.0%) (Appendix A).

### 3.4. Barriers to HZ Vaccination

Among the 272 participants who had not received any HZ vaccinations despite being aware of HZ vaccines, 87 (32.0%) reported in Part 1 of the survey (pre-intervention) that they were “very” interested in receiving the HZ vaccine. Among these individuals, a broad range of reasons for not getting the vaccine were reported, with cost (*n* = 16, 18.4%), not being recommended by a physician/nurse (*n* = 15, 17.2%), and concerns about side effects (*n* = 13, 14.9%) most frequently cited. For the 185 (68.0%) participants who were unvaccinated and undecided, not interested, or only “somewhat” interested in vaccination, the most frequently reported reasons were concern about side effects (*n* = 71, 38.4%), no vaccine recommendation from a physician or nurse (*n* = 67, 36.2%), not knowing enough about the vaccine (*n* = 58, 31.4%), concern it could cause HZ (*n* = 41, 22.2%), and that “no one else” had recommended it (*n* = 33, 17.8%) (Appendix A). The recent COVID-19 pandemic has been reported to impact overall vaccine receipt and hesitancy [36], but almost all participants in this study reported that the pandemic had not changed (*n* = 543, 73.9%) or had increased (*n* = 169, 23.0%) their interest in receiving HZ vaccination; only 23 (3.1%) participants stated that the pandemic had decreased their interest.

### 3.5. Responses after the HZ Educational Video

A total of 492 participants were eligible for the intervention, of whom 476 (96.7%) watched the video and completed Part 2 of the survey (Figure 1). Over 90% of Part 2 participants considered all the information in the video to be useful with 70% reporting it to be very useful. The participants considered the most useful information in the video to be that regarding the increased risk of HZ due to COPD (89.0% of participants) and general information on the risk and burden of HZ (80.7% of participants). Of the 476 participants, only 129 (27.1%) had reported in Part 1 that they knew that COPD increased HZ risk. Among the other 347 (72.9%) participants who did not know that COPD increased HZ risk, 311 (89.6%) reported after watching the video that they now believed that COPD increased HZ risk (Figure 3A). After watching the video, 208 (78.8%) of the 264 Part 2 respondents who knew of HZ vaccines but had not received any HZ vaccine reported that they now had intent to either get the HZ vaccine or discuss the HZ vaccine with their HCP. As anticipated, those reporting they were very interested before watching the video (*n* = 85) were most likely to report post-video intent to get the vaccine (80.0%) or speak with their HCP about the vaccine (14.0%) compared with those with lower levels of pre-video interest. However, even among those stating they were not at all interested in getting the HZ vaccine (*n* = 78), undecided (*n* = 67), or somewhat interested (*n* = 78) before the video, 29.4%, 73.1%, and 88.5% respectively now reported intent to either get the vaccine or speak with their HCP about the vaccine, while 55.9%, 9.0%, and 2.6% respectively reported no intent or plan to get the HZ vaccine (Figure 3A,B).

When considering the potential effect of demographic factors on the post-video vaccine intent versus pre-vaccine level of interest using an ordinal intent scale from “no plans to get vaccine” to “will consider getting vaccine” to “will speak with HCP about HZ vaccine” to “plan to get the HZ vaccine”, age, gender, COPD exacerbation history, and overall health all had significant effects. Regarding age, younger individuals were less likely to move up the scale (estimate = −0.041), women were more likely to move up the scale (estimate 0.402), and those with any COPD exacerbation in the prior year were more likely to move up the scale (estimate 0.146), as were those with worse overall health (estimate 0.140), *p* < 0.001. The ordinal association measure gamma, used to assess the size of the effect across all levels of pre-video interest and post-video intent, had a value of 0.369 (*p* < 0.001); when calculated excluding those “very interested” pre-video, the gamma value was 0.290. Both gamma values were close to or greater than 0.30 (*p* < 0.037) and therefore considered a moderate to strong effect size.

Using the Bonferroni correction did not change the statistical significance of any of the outcomes from univariate post-hoc analyses.

## 4. Discussion

Among this cohort of adults with COPD, awareness of HZ was almost universal, but three-quarters were unaware that having COPD is associated with an increased risk of developing HZ. While 57.0% of participants reported the receipt of some form of HZ vaccine, only 33.1% could confirm receipt of ACIP-preferred HZ vaccinations [23]. Among those who had experienced HZ and those with family or friends who had experienced HZ, the acute burden of pain and rash was commonly recognized and reported as important to prevent. Among participants who had experienced HZ, about one in four reported an increase in COPD symptoms, dyspnea, or exacerbation, or “flare”, associated with the acute HZ episode. The response to the educational video (the intervention) was positive, with a substantial increase in both awareness that COPD increases HZ risk and a reported interest in receiving vaccinations for HZ prevention.

In this cohort, the 192 individuals with prior HZ episodes reported rates of HZ symptoms and impact similar to those reported in the literature; most participants reported pain/itching, rash, and some interruption of usual activities, 15.6% of participants reported PHN, and 7.3% reported HZ eye complications [5,8,37]. What has received limited attention in previous studies [15,19], and for which we found no published reports from a USA population, was the reported increase in COPD-related symptoms (18.8%), dyspnea (25.5%), and exacerbations (12.0%) that participants associated with their acute HZ episode. This may reflect an increase in acute HZ episodes during COPD exacerbations or, alternatively, increased COPD symptoms following VZV reactivation. An increase in COPD-related symptoms with HZ was more commonly reported among those who reported worse overall physical and mental health. Further investigation is warranted to confirm our finding of increased COPD symptoms with HZ and to understand the underlying pathophysiology. If confirmed, this association could provide additional impetus to recommend and provide vaccine protection against HZ in people living with COPD.

Recent studies have demonstrated an increased rate of HZ in people with several chronic conditions including COPD [11,14,15,16,19]. However, this group of people with COPD were mostly unaware of this risk, and some even reported that COPD might decrease the risk of HZ. Considering the reported burden of HZ, such as pain and impact on usual activities and COPD-related symptoms, among those who had experienced HZ with their COPD or reported knowing family or friends with HZ (with or without concomitant COPD), knowing that they are at increased risk may aid their decision to receive HZ prevention.

Most reports of HZ vaccine uptake include receipt of any HZ vaccine [25,26]. Following the ACIP’s preferential recommendation for the recombinant vaccine in adults aged 50 years and older, including those aged 60 years and older who have received the inactivated HZ vaccine [23], plus confirmation of the waning of vaccine-induced immunity over 3–5 years with the older HZ vaccine [38,39], it is important to report not just HZ vaccine uptake but ACIP-preferred HZ vaccine uptake. This would more accurately reflect rates of longer-term prevention and might help persuade more people with COPD to get vaccinated and their HCPs to provide, or at least recommend, the ACIP-preferred HZ prevention strategy.

Barriers to vaccine uptake, including vaccine hesitancy, have been widely reported for the general public and for older adults [28,29]. Of particular note in this group of adults with COPD (who are sufficiently engaged to join a COPD registry) is the limited number who remembered HZ vaccination recommendations by their HCP. The marked differences between primary care (68.8%), pharmacy (48.5%), and pulmonologists (26.6%) cannot be explained by differences in rates of visiting those sites since the participants were asked to mark “does not apply or I don’t go there” for sites they do not attend. While HCPs may have a different focus for their visits, the rates of increased HZ-related COPD symptoms and exacerbations suggest the importance of HZ vaccine recommendations by all HCPs. Among participants who had not received any HZ vaccinations and were undecided, not interested, or only somewhat interested in vaccination, the second most common reason for not getting vaccinated was that their physician or nurse had not recommended it (36.2%), while a further 17.8% reported that “no one else” had recommended it. The recall of HZ recommendations was much less frequent (lower) than participants had reported regarding HCP vaccine discussions or recommendations for “pneumonia” and influenza vaccines, suggesting that HCPs could add HZ to other commonly recommended vaccines. It is also interesting that over half of participants reported that they remembered a discussion of HZ vaccinations with family and friends or through media, a higher proportion than from either pharmacy or pulmonologists. 

In this interventional survey study, we were able to show a measurable change in the rate of HZ awareness as related to risk in individuals with COPD, with this increasing from 27.1% to 89.6% through educational video intervention. Those who were not vaccinated according to the ACIP-preferred vaccine strategy also reported a measurable increase in intent to receive the HZ vaccine or discuss it with their HCP using paired analyses. The effect size for the video was “strong” with a gamma measure of 0.369. While those respondents most interested pre-video had the highest levels of vaccine intent, even those not at all interested pre-video moved towards intent to get the vaccine or speak with an HCP in about 30% of individuals. While reports of intent are not the same as actual vaccination uptake rate changes, they support the potential utility of a simple and relatively inexpensive online educational video to increase awareness and vaccine intent. Similar short (5-min) video programs might be useful in HCP waiting rooms, through sharing on patient portals, or other HCP or health system communication mechanisms.

This study has both strengths and weaknesses. Using an engaged cohort of individuals with COPD with online access allowed the survey to be longer, to incorporate skip patterns to limit completion time, and to embed an educational intervention, e.g., a video for a select subgroup. The response rate was not high but was comparable to many “patient surveys” [40,41], especially during the COVID-19 pandemic when patients in registries report receiving many surveys. The participants are not representative of all people with COPD in the USA but have similar demographic characteristics to the entire registry. Having a group engaged in their own health and with high rates of other recommended vaccines, specifically pneumococcal and influenza vaccines, allowed some differentiation between issues for HZ vaccine uptake and uptake of other common adult vaccines. That COPD diagnosis and vaccination status were self-reported may also be considered limitations. However, patient reporting is very useful for reporting remembered prior recommendations, assessing illness burden, measuring the perceived importance of disease prevention, and vaccine barriers, which cannot be obtained through medical records or administrative database studies [42]. As with most online surveys, it is not possible to confirm that the person answering the survey is the person to whom the invitation was sent. However, our use of a cohort who self-registered for the COPD PPRN and consented to receive research opportunities likely increases the chances that the respondent was the individual to whom the email was sent and who had provided prior demographic and COPD disease-related data. Our choice to use slightly different pre-video and post-video questions to move from the level of vaccine interest pre-video to an action—the level of vaccine intent—post video might be considered a limitation. 

## 5. Conclusions

The responses of this cohort of engaged members of a self-reported COPD registry highlight the limited awareness of the increased risk of HZ in those with COPD and confirm the significant perceived burden of acute HZ, its associated pain, and its impact on daily life. Of unique interest is the perceived HZ-associated increase in COPD-related symptoms and dyspnea in up to one-quarter of those who experienced a HZ episode concomitant with their COPD. This study highlights multiple opportunities to potentially improve patient awareness and vaccine intent through a short educational video, reveals the limited patient experiences of HCP recommendations for HZ prevention, and the limitations of current reports that do not include ACIP-preferred HZ vaccination rates.

## Figures and Tables

**Figure 1 vaccines-10-00420-f001:**
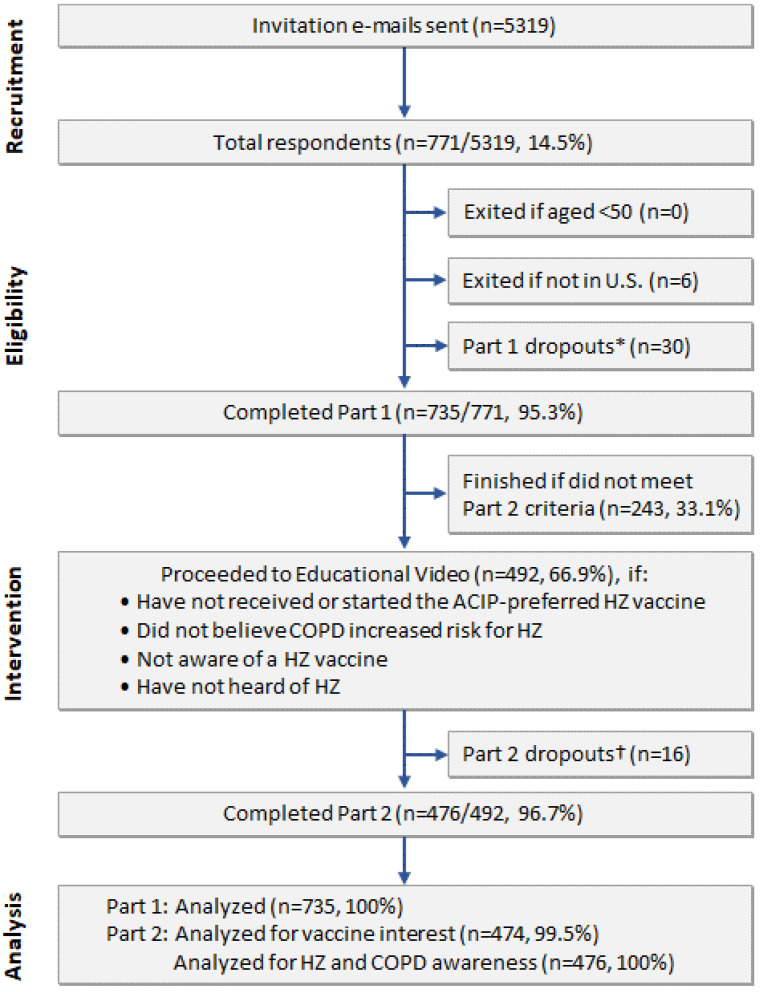
Flow diagram of participant disposition in the Shingles Patient Prevention Study (ShiPPS). * Participants who started Part 1 but stopped at various points before completing Part 1. ^†^ Participants who were not excluded but did not start Part 2. Abbreviations: ACIP, Advisory Committee on Immunization Practices; COPD, chronic obstructive pulmonary disease; HZ, herpes zoster; U.S., United States.

**Figure 2 vaccines-10-00420-f002:**
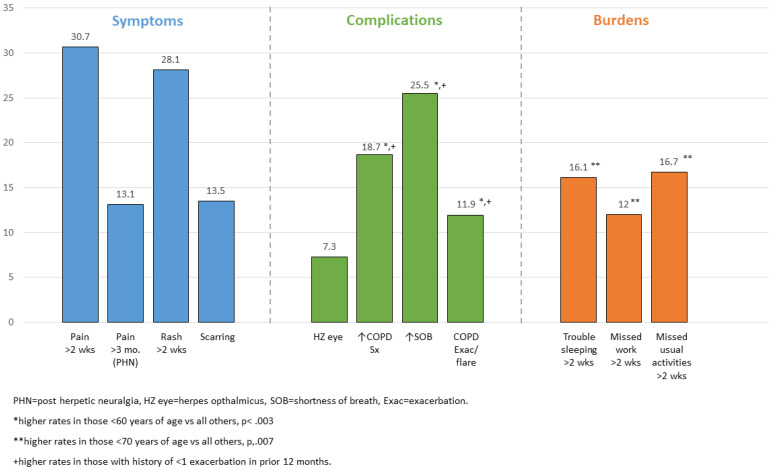
Long term HZ symptoms, complications, and burden experienced by participants who had previously had HZ. (% of *n* = 192).

**Figure 3 vaccines-10-00420-f003:**
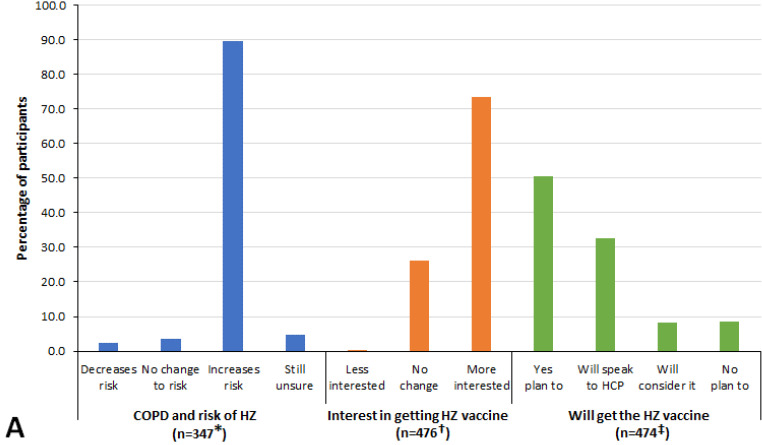
Changes after watching the educational video in (**A**) participants’ understanding of HZ risk and intent to get vaccinated, and (**B**) vaccine intent post-video versus level of vaccine interest pre-video in those not yet vaccinated. * Participants who reported before watching the video that they did not think that COPD increased the risk of HZ. ^†^ Participants who completed Part 2 of the survey. ^‡^ The two participants who answered that they were less interested in getting the HZ vaccine were not asked whether they now planned to get the vaccine (per the survey skip pattern). *Abbreviations:* COPD, chronic obstructive pulmonary disease; HCP, healthcare professional; HZ, herpes zoster.

**Table 1 vaccines-10-00420-t001:** Demographics and clinical characteristics for participants with COPD.

Characteristic	Participants with COPD (*n* = 735)
Age, mean (SD), years	68.5 (7.9)
Male/Female, *n* (%) *	296/438 (40.3/59.6)
Ethnicity, *n* (%)	
Caucasian	688 (93.6)
Black or African American	18 (2.4)
Multi-ethnicity	15 (2.0)
Other/Unknown	14 (1.9)
Overall health, *n* (%)	
Very good or excellent	63 (8.6)
Good	283 (38.5)
Poor or fair	389 (52.9)
Number of days in the past 30 days where physical health was not good, mean (SD)	10.7 (11.0)
Number of days in the past 30 days where mental health was not good, mean (SD)	6.8 (8.8)
Number of days in the past 30 days where poor physical or mental health prevented usual activity, mean (SD)	8.9 (10.3)
One or more exacerbations in the prior year, *n* (%) ^†^	395 (53.7)
CAT score, mean (SD)	19.6 (8.0)
Previously had HZ, *n* (%) ^‡^	192 (26.1)
Know others who have had HZ, *n* (%) ^‡^	492 (66.9)
Previously vaccinated against HZ, *n* (%) ^§^	419 (57.0)
One-shot vaccine only (Zostavax^®^ (Merck, Kenilworth, NJ, USA))	116 (15.8)
Two-shot vaccine (Shingrix^®^ (GlaxoSmithKline, Phildelphia, PA, USA); alone or plus Zostavax^®^)	243 (33.1)
Not known which kind	60 (8.2)

* A preference not to answer was given by one participant. ^†^ Exacerbation or worsening of COPD symptoms requiring treatment with antibiotics or steroids. ^‡^
*n* = 706; only includes those who had heard of HZ. ^§^
*n* = 691; only includes those who had heard of HZ vaccine. Abbreviations: CAT, COPD assessment test; COPD, chronic obstructive pulmonary disease; HZ, herpes zoster; SD, standard deviation.

**Table 2 vaccines-10-00420-t002:** Participants’ perceived impact of symptoms and problems of HZ, and the importance of preventing them, for people with COPD.

Symptom/Problem	How Bothersome Are the Symptoms and Problems for People with COPD, *n* (%) (*n* = 735) *			How Important Is Prevention of the Symptoms and Problems for People with COPD, *n* (%) (*n* = 735) *		
	Not at All	Somewhat	Very	Not at All	Somewhat	Very
HZ rash	19 (2.6)	169 (23.0)	547 (74.4)	11 (1.5)	91 (12.4)	633 (86.1)
HZ pain: 1–4 weeks	9 (1.2)	100 (13.6)	626 (85.2)	4 (0.5)	64 (8.7)	667 (90.7)
1–3 months	18 (2.4)	66 (9.0)	651 (88.6)	3 (0.4)	46 (6.3)	686 (93.3)
>3 months	27 (3.7)	58 (7.9)	650 (88.4)	6 (0.8)	39 (5.3)	690 (93.9)
Fear of infecting others	110 (15.0)	216 (29.4)	409 (55.6)	62 (8.4)	130 (17.7)	543 (73.9)
Persistent sadness or anxiety	69 (9.4)	251 (34.1)	415 (56.5)	33 (4.5)	142 (19.3)	560 (76.2)
Missing work or usual activities	51 (6.9)	206 (28.0)	478 (65.0)	29 (3.9)	131 (17.9)	575 (78.2)
Missed social activities	54 (7.3)	222 (30.2)	459 (62.4)	35 (4.8)	171 (23.3)	529 (72.0)
Increased COPD symptoms	64 (8.7)	148 (20.1)	523 (71.2)	25 (3.4)	89 (12.1)	621 (84.5)
COPD exacerbation	66 (9.0)	134 (18.2)	535 (72.8)	28 (3.8)	72 (9.8)	635 (86.4)

* Original scoring of responses used a scale of 1 to 5, where 1 was ‘not at all’ bothersome/important and 5 was ‘very’ bothersome/important. Scores of 2 and 3 were combined under the ‘somewhat’ response category, and scores of 4 and 5 were combined under the ‘very’ response. Abbreviations: COPD, chronic obstructive pulmonary disease; HZ, herpes zoster.

## Data Availability

The data presented in this study are available on request from the corresponding author. The data are not publicly available due to ongoing analyses of the dataset.

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
