# Peer review of "Knowledge and Attitudes Concerning Herpes Zoster among People with COPD: An Interventional Survey Study"

_vaccines, 2022, doi:10.3390/vaccines10030420_

Round 1

Reviewer 1 Report

Minor comment:

1. The entire study (survey) was conducted on-line. How did the authors verify who exactly watched the movie and who exactly filled in the questionnaire? The question is whether these were the patients to whom they send an invitation whether someone else did it for them. It is very important because it determines the final result of the survey. If this is was not confirmed, it should be mentioned as very important limitation. 

Author Response

Point---how do you know that the person who responded was the person to whom the email was addressed.

Response---As with any online survey, we cannot be certain that the person responding was the person to whom the email was sent.  It is possible that another individual responded and answered questions about COPD and history of HZ and HZ vaccination for the individual who was queried.  The fact that we used a registry for which patients had signed themselves up to receive research surveys and provided baseline and often yearly updated data may make this concern less likely than for a survey that used lists of individuals who were not part of a patient registry for which the patients had consented to receive research opportunities.  We have added sentences to the limitations to say that this is a possible limitation that is not quantifiable.

As with most online surveys, it is not possible to confirm that the person answering the survey is the person to whom the invitation was sent. However, our use of a cohort who self-registered for The COPD PPRN and consented to receive research opportunities likely increases the chances that the respondent was the individual to whom the email was sent and who had provided prior demographic and COPD disease-related data.

Reviewer 2 Report

Authors addressed most of my comments. However, it is not true that multiple comparison adjustment in case of a sequence of univariate analyses applies only when the same outcome (or family of outcomes) is considered. I would strongly encourage to either apply this adjustment or to consider it as a major limitation to address, providing reasons for this choice 

Author Response

Point---Bonferronie or other corrections should be used when making mulitple univariate comparisons.  This should be done or listed as a major limitation.

Response--We have extended the analyses to include use of Bonferroni correction for all post hoc univariate analyses with multiple comparisons.  Since almost all of these had p values <.001 to <.0001, the correction did not alter the reported statistical significance in any outcome.  We have added sentences to the methods section and at the end of the results section.

In methods section---Bonferroni correction was used to address multiple comparisons for all post-hoc univariate analyses.

At end of results section---Using the Bonferroni correction did not change the statistical significance of any of the outcomes from univariate post-hoc analyses.

This manuscript is a resubmission of an earlier submission. The following is a list of the peer review reports and author responses from that submission.

Round 1

Reviewer 1 Report

This is an interesting study, statistical analysis however is inappropriate. In contrast to the Authors conclusions the impact of the educational video was actually not observed. Figure 3B shows that among Undecided (before watching video) 60% remained undecided (i.e. Will speak to HCP), among Not at all interested almost 60% remained not at all interested (No plan to), among Somewhat interested - the same situation and among Very interested 80% was steel Very interested (i.e. Yes, plan to). This shows that patients interested in vaccination before video remained convinced after watching the video, whereas patients not at all interested before, still remained not interested at all even watching the video.

Second problem is that the answers after watching the video were not the same as the answers before, whereas they should be. This is a serious mistake in the study design.

The third problem - statistical analysis. In case of Figure 3B logistic regression with multinomial ordinal responses should be used to model effect of educational intervention and other important variables should be incorporate into the model to adjust obtained results. In such model X-values on the Fig. 3B will be ordinal independent variable. Dependent variable has four levels (also ordinal): Yes plan to, Will speak to HCP, Will consider it and No plan to.

In the case of Fig. 3B an ordinal association measure gamma should be used to obtain the size of effect of educational intervention.

Author Response

Reviewer 1:

Issue 1.  Analyses were inappropriate.  Figure 3B--shows that patients interested in vaccination before video remained convinced after watching the video, whereas patients not at all interested before, still remained not interested at all even watching the video.

Thank you for this suggestion.   While your interpretation by just looking is useful, the analyses suggest that the video did have a significant effect when assessing the paired responses for only those that were not initially very interested (ordinal association measure gamma = .0369, p<.001) which is greater than a gamma of .30 which is considered to a strong effect.

We do disagree that the group who were very interested are of no interest.  While this is only intent, they moved from very interested to stated intent to obtain vaccine which is a potentially important change.  

Issue 2. Second problem is that the answers after watching the video were not the same as the answers before, whereas they should be. This is a serious mistake in the study design.

Thank you for this observation.  We believe that it is not necessary to have exactly the same answers before and after the video since we included only a subset of the full population in the after questions---those who had not been vaccinated. Some individuals who develop questionnaires feel the answers must be comparable for comparison but need not be identical when the video or other intervention may present additional options as we feel we did.  Obviously, this could not be changed even if we felt it was a mistake in study design.

Issue 3. The third problem - statistical analysis. In case of Figure 3B logistic regression with multinomial ordinal responses should be used to model effect of educational intervention and other important variables should be incorporate into the model to adjust obtained results. In such model X-values on the Fig. 3B will be ordinal independent variable. Dependent variable has four levels (also ordinal): Yes plan to, Will speak to HCP, Will consider it and No plan to.

Thank you for your suggestion.  We have completed multinomial ordinal regression with logit link function for the paired responses for the groups other than the very interested who watched the video.  As noted in the added text

When considering the potential effect of demographic factors on the change in vaccine interest on an ordinal scale from not interested or won’t get to will consider to will speak with HCP to plan to get, age, gender, COPD exacerbation history and overall health all have significant effects.  For age, younger individuals were less likely to move up the scale (estimate = -.041), women were more likely to move up the scale (estimate .402), those with any COPD exacerbation in the prior year were more likely to move up the scale (estimate .146) as were those with worse overall health (estimate .140)  All analyses were associated with p<.001).

Issue 4. In the case of Fig. 3B an ordinal association measure gamma should be used to obtain the size of effect of educational intervention.

Thank you, we did this assessment. 

“The ordinal association measure gamma used to assess the size of the effect was gamma = .369 (P<.001) which is greater than .30 and therefore considered a strong effect size.”

Reviewer 2 Report

Yawn and colleagues conducted a cross sectional study using data from a US registry of people with COPD to investigate HZ risk awareness, HZ vaccine use, and barriers in people with the condition. They also evaluated the change in vaccine intent after watching an educational video about HZ vaccine but I would highly discourage the use of the word "impact" given that the employed methodology does not support causal inference. 

Overall the study covers a relevant topic but I believe there are major points to address before considering it for publication. Please see my points below:

  • Author briefly described in the limitations that the study population is not representative of the people with COPD in the US but this point should be better explained also in the method section. Although a valid source of information, there are many issues with extracting data from registries as selection biases arise in the likelihood of participating onto the registry. Hence, authors should clearly describe this aspect in the method section as well.
  • In line with the previous point, authors should elaborate on whether the study population differs from the registry population, considering the very low participation rate. I wonder why they did not compare differences between participants and non-participants
  • Authors employed simple descriptive statistics. Whilst I might partially understand the reasons behind this choice I wonder why they did not employ more complex statistics which would have provided more accurate estimates given the relatively large sample. Given they decided to stick with descriptive statistics, they should at least consider correcting results for multiple comparisons.
  • To describe frequency of symptoms occurrence authors used a spider plot. I wonder why they did not consider using more sophisticated approaches (i.e. latent class analysis) to profile symptoms clustering
  • Did authors employ paired statistics for the second part of the study evaluating the effect of the video intervention? They should as the before and after answers should be clustered within the same individual

Author Response

Reviewer 2

Issue 1. Yawn and colleagues conducted a cross sectional study using data from a US registry of people with COPD to investigate HZ risk awareness, HZ vaccine use, and barriers in people with the condition. They also evaluated the change in vaccine intent after watching an educational video about HZ vaccine but I would highly discourage the use of the word "impact" given that the employed methodology does not support causal inference. 

We agree that we are not claiming casual inference.  We have therefore edited the sentence that used the term“impact”in the discussion section.

In this interventional survey study, we were able to show a measurable change on the rate of HZ awareness as related to risk in individuals with COPD…..”

Also edited the title to Figure 3.

Figure 3. Changes after watching the educational video for (A) participants’ understanding of HZ risk and interest in and plans to get vaccinated, and (B) plans……

Issue 2. Author briefly described in the limitations that the study population is not representative of the people with COPD in the US but this point should be better explained also in the method section. Although a valid source of information, there are many issues with extracting data from registries as selection biases arise in the likelihood of participating onto the registry. Hence, authors should clearly describe this aspect in the method section as well.

Thank you for the suggestion.  Due to word and space limitations, we did not comment on the limitations of using a registry in the methods section.  However, we have now added two sentences to highlight registry limitations in the methods section. 

Registry populations may not be representative the population of interest since most require active choices to participate and this registry requires access to online registration.”

In addition, we have added some general demographic information in the results section to allow comparisons with the total registry population.

 These demographic characteristics are very similar to those for the entire COPD PPRN population: mean age 68.1 (s.d. 8.8) years, 60.1% females and 50.3% reported exacerbations in the prior 12 months”

Issue 3. In line with the previous point, authors should elaborate on whether the study population differs from the registry population, considering the very low participation rate. I wonder why they did not compare differences between participants and non-participants.

Thank you for this suggestion.  As noted above, we have added information related to age,  gender and exacerbation distribution of the respondents and the entire registry population. 

Issue 4. Authors employed simple descriptive statistics. Whilst I might partially understand the reasons behind this choice I wonder why they did not employ more complex statistics which would have provided more accurate estimates given the relatively large sample. Given they decided to stick with descriptive statistics, they should at least consider correcting results for multiple comparisons.

We considered such adjustments.  However, statistical tests were selected based on literature and understanding of patients with COPD and herpes zoster. We believe that correcting for multiple comparisons would be needed if we were comparing the same variables, however, that is not the case in this paper.

Issue 5. To describe frequency of symptoms occurrence authors used a spider plot. I wonder why they did not consider using more sophisticated approaches (i.e. latent class analysis) to profile symptoms clustering

We agree that the spider plot, while interesting in appearance, does not really highlight the data clearly.  We have deleted the spider plot and substituted a histogram that highlights the issues of longer duration and those specifically associated with COPD which we have not found reported elsewhere.  Symptom clustering does not appear to be appropriate for this data or for HZ since pain and rash are almost universal and complications have not been shown to be “clustered”.  We did add comments on associations of patient demographics and COPD associated symptoms/burdens.

Issue 6. Did authors employ paired statistics for the second part of the study evaluating the effect of the video intervention? They should as the before and after answers should be clustered within the same individual.

Thank you for the suggestion.  We have added paired statistics to our analyses and added the outcome of the analyses to the results section and expanded comments in the discussion.  Please note that we believe it is important to leave the group that was initially “interested” but had not yet been vaccinated since the outcome we hope will be achieved is receipt of vaccination.

Please see the responses to Reviewer 1.

Round 2

Reviewer 1 Report

Regarding to: Analyses were inappropriate.  Figure 3B--shows that patients interested in vaccination before video remained convinced after watching the video, whereas patients not at all interested before, still remained not interested at all even watching the video.

With regret but in my opinion your explanation is not convincing and results are still far-fetched.

Regarding to: Second problem is that the answers after watching the video were not the same as the answers before, whereas they should be. This is a serious mistake in the study design.

Unfortunatelly, I still stand by my opinion that this is important mistake in "before-after" study. The test conditions should be exactly the same.

Reviewer 2 Report

I have not additional comments